# Automatic Deep-Learning Segmentation of Epicardial Adipose Tissue from Low-Dose Chest CT and Prognosis Impact on COVID-19

**DOI:** 10.3390/cells11061034

**Published:** 2022-03-18

**Authors:** Axel Bartoli, Joris Fournel, Léa Ait-Yahia, Farah Cadour, Farouk Tradi, Badih Ghattas, Sébastien Cortaredona, Matthieu Million, Adèle Lasbleiz, Anne Dutour, Bénédicte Gaborit, Alexis Jacquier

**Affiliations:** 1Department of Radiology, Hôpital de la TIMONE, AP-HM, 13005 Marseille, France; lea.ait-yahia@ap-hm.fr (L.A.-Y.); farah.cadour@ap-hm.fr (F.C.); farouk.tradi@ap-hm.fr (F.T.); alexis.jacquier@ap-hm.fr (A.J.); 2CRMBM—UMR CNRS 7339, Aix-Marseille University, 27, Boulevard Jean Moulin, 13005 Marseille, France; jorisfournell@gmail.com; 3I2M—UMR CNRS 7373, Luminy Faculty of Sciences, Aix-Marseille University, 163 Avenue de Luminy, Case 901, 13009 Marseille, France; badihghattas@gmail.com; 4IHU Méditerranée Infection, 19–21 Boulevard Jean Moulin, 13005 Marseille, France; sebastien.cortaredona@inserm.fr (S.C.); matthieu.million@ap-hm.fr (M.M.); 5VITROME, SSA, IRD, Aix-Marseille University, 13005 Marseille, France; 6MEPHI, IRD, AP-HM, Aix Marseille University, 13005 Marseille, France; 7C2VN, INRAE, INSERM, Aix Marseille University, 27, Boulevard Jean Moulin, 13005 Marseille, France; adele.lasbleiz@ap-hm.fr (A.L.); anne.dutour@ap-hm.fr (A.D.); benedicte.gaborit@ap-hm.fr (B.G.); 8Department of Endocrinology, Metabolic Diseases and Nutrition, Pôle ENDO, AP-HM, 13915 Marseille, France

**Keywords:** adipose tissue, thoracic imaging, artificial intelligence, deep-learning, COVID-19

## Abstract

Background: To develop a deep-learning (DL) pipeline that allowed an automated segmentation of epicardial adipose tissue (EAT) from low-dose computed tomography (LDCT) and investigate the link between EAT and COVID-19 clinical outcomes. Methods: This monocentric retrospective study included 353 patients: 95 for training, 20 for testing, and 238 for prognosis evaluation. EAT segmentation was obtained after thresholding on a manually segmented pericardial volume. The model was evaluated with Dice coefficient (DSC), inter-and intraobserver reproducibility, and clinical measures. Uni-and multi-variate analyzes were conducted to assess the prognosis value of the EAT volume, EAT extent, and lung lesion extent on clinical outcomes, including hospitalization, oxygen therapy, intensive care unit admission and death. Results: The mean DSC for EAT volumes was 0.85 ± 0.05. For EAT volume, the mean absolute error was 11.7 ± 8.1 cm^3^ with a non-significant bias of −4.0 ± 13.9 cm^3^ and a correlation of 0.963 with the manual measures (*p* < 0.01). The multivariate model providing the higher AUC to predict adverse outcome include both EAT extent and lung lesion extent (AUC = 0.805). Conclusions: A DL algorithm was developed and evaluated to obtain reproducible and precise EAT segmentation on LDCT. EAT extent in association with lung lesion extent was associated with adverse clinical outcomes with an AUC = 0.805.

## 1. Introduction

Epicardial adipose tissue (EAT) is a unique adipose tissue that surrounds the heart and is located between the myocardium and the visceral layer of the pericardium [1]. The role of EAT with regard to the heart can be generally distinguished by mechanical, metabolic, thermogenic, and endocrine/paracrine functions [2,3]. A major volume of EAT is linked to an increased amount of fatal and non-fatal cardiovascular events [4], coronary calcification [5], carotid stiffness [6] and atrial fibrillation [7]. EAT volume is now considered a potential therapeutic target for cardiovascular disease, although it is not routinely measured [8]. Consequently, its role has been suggested in COVID-19 infections, because inflammation plays a major role in the progression of severe COVID-19 infection [9,10,11]. EAT has recently been associated with pneumonia lesions, myocardial damage and adverse outcomes in COVID-19 [12,13,14]. However, other studies did not find such an association [15,16]. In particular, in a post-hoc analysis including 192 patients, Conte et al. showed that EAT attenuation but not obesity or EAT volume was significantly associated with the risk of ICU admission, death or invasive ventilation in COVID-19 disease [17]. The preliminary step in these studies is to segment both COVID-19 lung lesion extent and EAT volume in clinical care with fast, reproducible techniques that are available. Recent segmentation studies have focused on deep-learning (DL) techniques, especially the use of convolutional neural networks (CNNs), which have shown promising results in the automation of medical imaging measures and segmentation [18]. Non-contrast computerized tomography (CT) has been described as a more sensitive and accurate tool for measuring epicardial fat thickness and volume, with high-resolution images that provide more details [19,20]. Low-dose chest CT (LDCT) is widely used in COVID-19 examinations.

Our main purpose was to develop and evaluate a complete DL pipeline that allows a fully automated segmentation of EAT volume that could be used on LDCT in association with a pre-existing COVID-19 lung lesions segmentation tool. We also aimed to investigate the association between EAT volume and lung lesion extent in a COVID-19 cohort and the impact of EAT volume on patient prognosis.

## 2. Materials and Methods

### 2.1. Study Design

This single-center retrospective study was conducted from January 2021 to March 2021. The conformity of the study with European data privacy rules was approved by the local data protection officer and complied with the Declaration of Helsinki (N°:20200012, RGPD/ApHm: 2020-48). A total of 353 consecutive patients with COVID-19 confirmed by acute respiratory syndrome coronavirus 2 (SARS-CoV-2) reverse transcription polymerase chain reaction (RT-PCR) test and who underwent LDCT for lesions extension were retrospectively selected for inclusion in the study. Some 115 patients were randomly selected to develop a complete pipeline based on a CNN to segment and quantify EAT in LDCT. A group of 95 patients were randomly assigned to the training dataset (47214 CT slices), and 20 patients were assigned to the testing dataset (10412 CT slices). An external cohort of 238 patients was selected to assess the correlation between lung lesions extent and EAT volume and the potential impact of EAT volume on clinical prognosis. Automatic segmentation of lung lesions due to COVID-19 was previously validated and used elsewhere [21]. A flow diagram of the procedure is shown in Figure 1.

### 2.2. Population and Data

#### 2.2.1. Population

All patients were enrolled from one single center (La Timone—Assistance Publique Hôpitaux de Marseille). All patients between 4 January and 30 March 2021, who had a positive RT-PCR test result for SARS-CoV-2 and underwent unenhanced LDCT were included. LDCT was performed on all patients over 55 years old or with risk factors for adverse outcomes of COVID-19, such as hypertension, diabetes, obesity (defined as a body mass index (BMI) ≥30 kg/m^2^), dyspnea, or abnormal lung auscultation. The exclusion criteria were protocol refusal or an age below 18 years. Patient follow-up lasted 10 days in cases of no adverse events, and the follow-up period was extended to cover an in-hospital stay for patients who required hospitalization. A need for oxygen therapy, intensive care unit (ICU) admission and death were recorded. All clinical data were anonymized.

#### 2.2.2. Radiological Data

All patients underwent unenhanced, deep-inspiration LDCT on the same system (Revolution EVO, General Electric Healthcare, Waukesha, WI, USA) with technical parameters detailed in Appendix A. The pre-established top anatomic border was the lower part of the neck. The bottom boundary was the location of the adrenal glands. All radiological data were strongly anonymized.

### 2.3. EAT Segmentation Model

#### 2.3.1. Manual Segmentation

Manual image segmentation was undertaken for the selected population by one observer: Observer 1 (O1); L.A.-Y., five years of experience. For each patient, images were extracted from the picture archiving and communicating system and imported in DICOM format on the validated post-processing software 3D Slicer (3D Slicer v4.11.20210226) [22]. Manual EAT segmentation and quantification were obtained in two phases on the mediastinal kernel [23]. First, manual segmentation was performed slice by slice on the entire intrapericardial soft tissue volume by delineating the external border of the pericardium using thresholding, painting, and erasing methods. The superior and inferior limits of the pericardium were first identified as the top of the left atrium, and the lower limit corresponded to the last slice in which the left ventricle was identified. Pericardial fat was excluded. This segmentation was named Peri. The EAT tissue was then identified inside the intrapericardial volume by using the standard fat attenuation range as a threshold, from −140 Hounsfield units (HU) to −30 HU [24]. The remaining voxel was considered the total volume of the EAT. EAT extent (Ext_EAT) was defined by the percentage of EAT volume on the Peri volume. The obtained segmentation masks were all validated by one experienced chest radiologist (A.J., 25 years of experience). All manual segmentations and extracted clinical measures were defined by O1a. The manual segmentation is presented in Figure 2.

#### 2.3.2. Manual Segmentation and Slice Detection Model

The classification model was obtained by training a multitask U-Net network in which the segmentation task would serve as a regularizer for the classification task. To this end, a modified U-Net network is trained where a classification head is added to the center of the U-Net in order to predict whether the current slice contains the heart. This network is trained with supervision signals from both the cross-entropy loss function for the classification task and from the segmentation loss function. When the training was finished, only the first part of the network was kept as our final slice detection network. The high-resolution network and object-context representation model specifications are presented in Appendix B. Algorithm development was run on a Biprocessor Intel Xeon Silver 4216 2.1GHz, RAM = 96Go, 2 GPU Nvidia Quadro RTX5000, 16Go.

#### 2.3.3. Performance Evaluation and Reproducibility

Manual Peri and EAT segmentations (O1a) were compared to the automatically obtained segmentations (Auto) on the testing dataset (n = 20). Technical performance was evaluated with the median and mean of the Sørensen-Dice coefficient (Dice). The O1a and Auto clinical parameters were evaluated in terms of segmentation volume (cm^3^) using mean absolute error (MAE), bias, and correlation. Efficiency, defined as the user interaction time comparison, was also evaluated. The reproducibility of the Auto method was compared to the inter-and intra-observer segmentation performances. Observer 1 performed a second analysis designated (O1b)., Observer 2 (O2; A.B., with six years of experience), performed 20 manual segmentations on the same testing dataset, designated (O2) and blind to the segmentations from O1.

### 2.4. Prognosis Value and Association with COVID-19 Lesions

To evaluate the association of EAT volume (EAT) and EAT extent (Ext_EAT) with COVID-19 lesions for prognosis, we used a cohort subset including 238 patients (Figure 1). The two developed segmentation tools were put in a row and performed both the EAT and lung lesion segmentations on the same LDCT. EAT segmentation was conducted on the mediastinal window, while the lung lesion segmentation was performed on the lung window. The lung lesion extent (Lesion_Ext, %) measures the percentage of lung lesions (ground-glass opacities and condensations) in the total lung volume. We studied the association between the EAT and lung lesion measures with four clinical outcomes: death, ICU transfer, need for oxygen therapy, and >10 days of hospitalization.

### 2.5. Statistical Analysis

The continuous and categorical variables are described by the mean, standard deviation (SD), range, and n (%). The Pearson correlation coefficient was used to measure pairwise linear dependence between lung lesions extent and epicardial adipose tissue volume. Student’s t-tests were performed to compare the means according to the clinical outcomes. Associations between the EAT volume or Ext_EAT and lung lesion measures and a combined clinical outcome (consisting of one or more of death, ICU transfer, need for oxygen therapy, and >10 days of hospitalization) were estimated using multivariable logistic regressions with adjustments for gender, age, and comorbidities (hypertension, diabetes, cancer, chronic respiratory disease, coronary artery disease, and obesity). The goodness of fit was assessed using the area under the receiver operating characteristic curve (AUC-ROC, or AUC) to measure the models’ ability to discriminate patients. A two-sided α value of less than 0.05 was considered statistically significant. The analyses were carried out using the SAS 9.4 statistical software (SAS Institute, Cary, NC, USA).

## 3. Results

### 3.1. Population Characteristics

A total of 353 patients were included; 115 were used to develop and test the EAT segmentation algorithm. The population characteristics are presented in Table 1. The EAT and Peri measures of the training and testing datasets were extracted from the manual O1a segmentations. EAT, Peri, and the COVID-19 lesions of the association dataset were extracted from the automatic model segmentations. The mean dose-length product (DLP) was 64.4 (±12.2) mGy cm2. The mean manual segmentation time for the EAT segmentation was 17.4 (±8.0) min versus <1 min for the automatic EAT segmentation (*p* < 0.0001).

### 3.2. EAT Algorithm Performance and Reproducibility

The results concerning the EAT segmentations in terms of the clinical and technical metrics on the test dataset are presented in Table 2. The mean Dice coefficients for the automatic segmentation of all the pericardial and EAT volumes were 0.93 ± 0.03 and 0.85 ± 0.05, respectively. For EAT, the median Dice reached 0.87. For the EAT volume measure, the MAE was 11.7 cm^3^ ± 8.1 with a non-significant measure bias of −4.0 cm^3^ ± 13.9 and a correlation of 0.963 with the manual measures (*p* < 0.01).

Concerning the algorithm performances in comparison to the inter-and intraobserver measure reproducibility (Table 2), the mean Dice for the EAT segmentation lesions was 0.85 ± 0.05 for O1a versus Auto, 0.85 ± 0.04 for O1a versus O1b, and 0.86 ± 0.03 for O1a versus O2. For the EAT measure, the MAE was 11.7 ± 8.1 with insignificant bias, while measured at 12.0 ± 9.1 for O1a versus O1b and 14.9 ± 11.2 for O1a versus O1b. The correlations between the automatic and manual Peri and EAT measures are presented in Figure 3. The Bland-Altman graphics are presented in Appendix C (Figure A1). 

### 3.3. EAT and COVID-19 Association

There was an association (Pearson correlation coefficient: 0.139; *p* = 0.037) between Lesion_Ext and EAT volume, as shown in Table 3 (and Appendix C—Figure A2). On the contrary, there was no statistically significant association between Lesion_Ext and EAT_Ext. The association between EAT volume and extent and the different related COVID-19 clinical outcomes, i.e., death, ICU transfer, need for oxygen therapy, and >10 days of hospitalization is presented in Appendix C (Table A1). EAT volume was significantly superior in patients with a need for oxygen therapy (125.3cm^3^ ± 53.9 versus 101.6 cm^3^ ± 61.9; *p* = 0.0023) and ICU admission (143.4cm^3^ ± 61.0 versus 108.3cm^3^ ± 57.8; *p* = 0.0023). There was no statistical difference in terms of Eat volume and Ext_EAT for hospitalizations of >10 days and death outcomes. Table 4 shows a significant link between the risk of an unfavorable clinical outcome and Ext_EAT (OR = 1.04; AUC = 0.744) and Lesion_Ext (OR = 1.10; AUC = 0.800). A model combining both the Ext_EAT and Lesion_Ext DL measures increased the model precision with AUC = 0.805 versus AUC = 0.733 (*p =* 0.0029) for the model without any DL measures. Appendix C shows all different models, including EAT volume and Ext_EAT (Table A2).

## 4. Discussion

The proposed automatic quantification pipeline provided an accurate, fast, and reproducible segmentation of EAT volume on LDCT. This allows the ability to obtain EAT volume on a clinical routine, with a non-invasive, accessible method and a low radiation dose, in less than 20 s.

Different methods have been proposed to develop EAT segmentation. Commandeur et al. proposed a multi-task CNN approach in which the network would jointly learn to segment the region inside the pericardium to classify each entry as belonging to epicardial fat, allowing them to train their model with all CT slices, regardless of the presence of cardiac tissue [25]. He et al. also used a pair of U-Nets combined sequentially [26,27]: the first U-Net model aimed at producing an initial segmentation of the region inside the pericardium, further refined by a morphological processing layer before being combined with the input image and sent to a second U-Net targeting the final EAT segmentation. Our approach is more conventional, as we found it more beneficial to distinguish the slice detection task and the segmentation task using two models instead of one as in [27].

We randomly selected a population from our daily routine activity to extrapolate our algorithm as much as possible to the general population. Notably, we chose to train our model on all patient morphotypes to be as representative as possible. Currently, there are no standard reference values for the measurement of epicardial fat volume or thickness in the general population and in patients with COVID-19 [13,28]. In addition, the accurate measurement of EAT volume and its relationship with body mass index (BMI) has not been fully elucidated [24]. In a recent systematic review and meta-analysis including more than 3500 patients by Nerlekar et al., the EAT volume was estimated to be about 108.5 cm^3^ (Min 76.4–Max140.6) in patients undergoing either intracoronary imaging or CT coronary angiography evaluation, and noninvasive measurement of EAT by either CT-derived volume or linear thickness [29]. Our results appeared to be lower in terms of the median value of the EAT volume. However, these patients all had a chronic or recent history of coronary artery disease (versus 11.58% in our training cohort) and a mean BMI of 29.4 kg/m^2^, which could explain the higher fat volume compared with our results. In the Heinz Nixdorf Recall study, which included 4093 participants free of cardiovascular disease from the general population, the mean EAT volume was measured at 85.9 cm^3^ (Min 61.4–Max120.9) [4,30]. Moreover, due to the anatomical variability of the pericardium between individuals, it sometimes appeared difficult to specify exactly the superior and inferior limits of the pericardium, and it can impact EAT volume measure.

Several automatic and semi-automatic software solutions have been proposed in the past years for EAT segmentation with different performances. Ding et al. proposed an automated segmentation model built on a geodesic active contours method with good correlation with manual measures for EAT volume (0.97) [30]. We obtained a similar correlation for the same measure (0.963), but segmentation time was shorter, less than 20 seconds versus one minute. Militello et al. obtained a higher Dice coefficient score of 0.93 versus 0.87 in our study for EAT volume [31]. In their study, EAT segmentation was conducted on cardiac CT with ECG synchronization. We trained our model on LDCT, with lower resolution and non-ECG synchronization, making it more complex to segment pericardium layers. We obtained good results on LDCT, which is widely used in clinical care, especially for COVID-19 examinations. EAT segmentation could be obtained on the same exam, without adding radiation dose. Commandeur et al. built a CNN model on a wider and multi-center population, but we obtained similar median Dice of about 0.87 [32]. Bias was non-significant and also similar on the EISNER trial, close to 4 cm^3^.

Association between visceral ectopic fat depots, EAT in particular, and COVID-19 has been a subject of major interest recently., Many studies did not show any association between EAT volume with pneumonia lung lesions extent while visceral fat was a marker of worse clinical outcomes [33]. We found a moderate but significant association between COVID-19 lesions extent and EAT volume. The hypothesis for this small association could be that the link between COVID-19 lesions and pericardial fat would mainly contribute to a local inflammatory effect on the neighboring lungs [34]. We did not study this correlation because the algorithm allows intrapericardial volume (Peri) and EAT segmentations, but not pericardial fat segmentation. Furthermore, EAT attenuation more than its amount could be associated with worse outcomes, as reported by Conte et al. [17]. However, recent studies have provided conflicting results [35]. The down regulation of ACE2 resulting from SARS-CoV-2 infection reduces this action due to reduced conversion of Ang II to Ang-(1–7) by ACE2, suggesting the possibility of EAT-mediated cardiac injury. Interestingly, it was also shown that diet-induced obesity led to greater ACE2 expression in EAT [36]. This raises the possibility for EAT in patients with obesity to serve as a hub for viral infections that could mediate infection of the heart [37]. Furthermore, the capability of EAT to release exosomes and microRNA that can enter cardiac cells open up numerous mechanisms by which EAT may contribute to/mediate the entry of the SARS-CoV2 into the heart, causing direct cardiac effects [38]. Grodecki et al. found a strong association between both measures, and a link between EAT and COVID-19 adverse outcomes [13,14]. Some of our results support these findings (Table A1), and EAT volume measurements could be integrated into the clinical evaluation of COVID-19 patients. The benefit of this algorithm is that it can be directly used on LDCT performed at the patient admission.

Our study has some limitations. All of the CT images studied were acquired on the same scanner in one clinical center. The use of different scanners and multicenter external validation are ongoing to validate the performance of the presented algorithm. The EAT/COVID-19 prognosis dataset included patients with a lower prevalence of diabetes, but multivariable analysis included adjustments for potential confounding factors such as diabetes and obesity. Another limitation in our study was the absence of patients who had a sternotomy, intracardiac device or a history of cardiac or pulmonary surgery. Although this represents a moderate part of the general population, we have not yet tested our algorithm in these different situations.

## 5. Conclusions

A complete deep-learning pipeline was developed for the segmentation of the EAT volume on LDCT acquisition. The automatically obtained radiological EAT volumes were precise and reproducible. EAT quantification can be obtained as a daily routine to evaluate cardiovascular and inflammatory risks. This inflammatory substrate is probably the reason why EAT volume is associated with more COVID-19 adverse outcomes, and further studies should be carried out to better understand this association.

## Figures and Tables

**Figure 1 cells-11-01034-f001:**
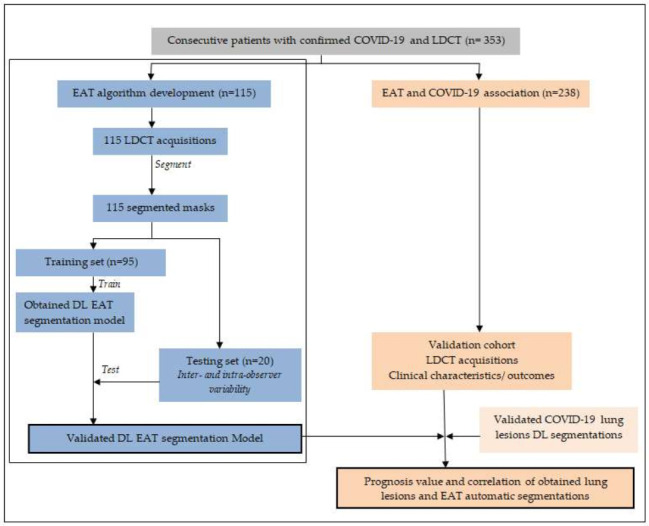
Study flow chart. LDCT: low-dose chest computed tomography; EAT: Epicardial adipose tissue pericardial volume; DL: Deep- Learning.

**Figure 2 cells-11-01034-f002:**
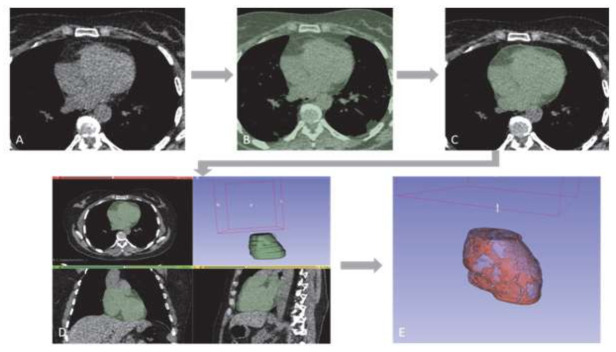
Example of manual segmentation of the Peri and EAT volumes in the LDCT. (**A**) LDCT axial image on mediastinal window; (**B**) mediastinal thresholding to exclude pulmonary parenchyma; (**C**) intrapericardial segmentation slice by slice using painting and erasing tools; (**D**) obtained 3D intrapericardial volume (Peri); (**E**) Obtained 3D EAT volume after standard fat attenuation thresholding.

**Figure 3 cells-11-01034-f003:**
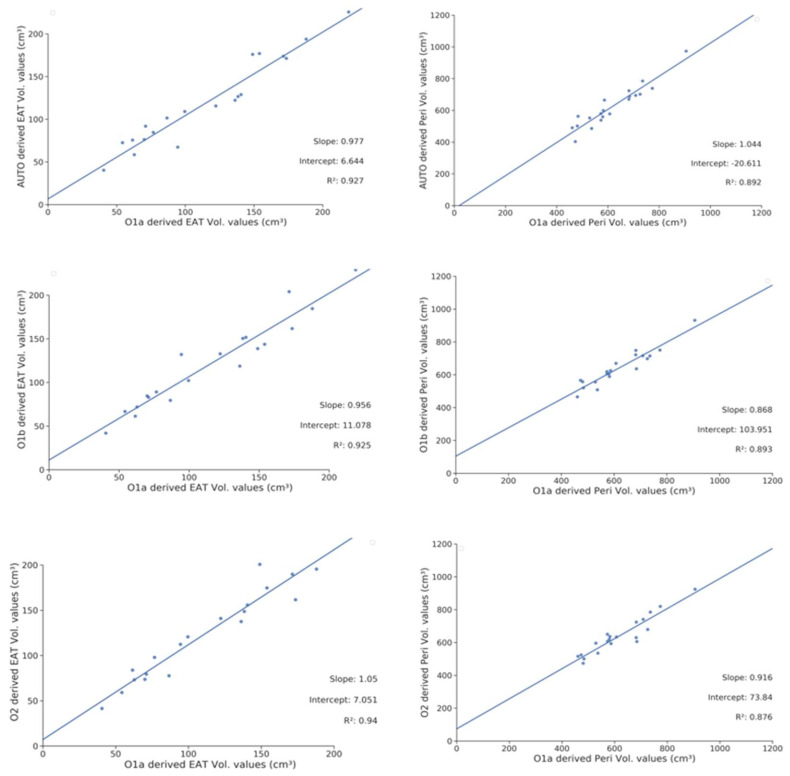
Correlations of EAT and Peri volume measures for the automatic and manual segmentation of the intra- and interobserver measures. Peri Vol: pericardial volume; EAT Vol: epicardial adipose tissue volume; O1a: initial segmentations made by Observer 1. O1b: second segmentations made by Observer 1; O2: Segmentation made by Observer 2. The blue line is the fitted regression line.

**Table 1 cells-11-01034-t001:** Population characteristics. SD: standard deviation; BSA: body surface index; Peri: intra-pericardial volume; EAT: epicardial adipose tissue volume; ICU: intensive care unit; Ext_EAT: epicardial adipose tissue extent (%); Lesion_Ext: COVID-19 lung lesion extent (%).

	EAT Segmentation Model Dataset		EAT/COVID-19 Prognosis Dataset
	Training Dataset (n = 95)	Testing Dataset (n = 20)	(n = 238)
Gender			
Male, n (%)	49 (51.58)	7 (35.00)	143 (60.01)
Age			
18–44 years, n (%)	16 (16.80)	4 (20.00)	32 (13.45)
45–64 years, n (%)	47 (49.47)	9 (45.00)	104 (43.70)
>64 years, n (%)	32 (33.68)	7 (35.00)	102 (42.86)
Body mass index (kg/m^2^), mean (±SD),	25.7 (±4.37)	24.3 (±4.28)	24.1 (±3.87)
Comorbidities			
Diabetes, n (%)	75 (78.94)	14 (70.00)	152 (63.4)
Hypertension, n (%)	46 (98.42)	11 (55.00)	97 (40.7)
Underweight, n (%)	0 (0.00)	0 (0.00)	3 (1.26)
Overweight, n (%)	24 (25.26)	7 (35.00)	88 (36.9)
Obesity, n (%)	18 (18.94)	3 (15.00)	47 (19.75)
Dyslipidemia, n (%)	28 (29.47)	6 (30.00)	62 (26.05)
Coronary artery disease, n (%)	11 (11.58)	1 (5.00)	44 (18.49)
Number of comorbidities			
None, n (%)	10 (10.52)	2 (10.00)	60 (25.21)
One, n (%)	27 (28.42)	5 (25.00)	80 (33.61)
Two or more, n (%)	58 (61.05)	13 (65.00)	98 (41.17)
Epicardial adipose tissue measures			
Peri (cm^3^), mean (±SD)	680.40 (±198.40)	617.93 (±104.47)	709.71 (±145.12)
Peri/BSA, (cm^3^/m^2^), mean	359.78	328.16	365.41
EAT (cm^3^), mean (±SD)	119.17 (±71.36)	115.47 (±49.12)	112.83 (±59.30)
EAT/BSA (cm^3^/m^2^), mean	63.05	61.32	58.83
Ext_EAT (%), mean (±SD)	17.51 (±21.64)	18.68 (±22.14)	15.60 (±6.50)
Delay symptoms—LDCT			
Delay <7 days/asymptomatic, n (%)	x	x	148 (62.18)
Delay ≥7 days, n (%)	x	x	90 (37.82)
COVID-19 pulmonary lesions			
Lesion _Ext (%), mean (± SD)	x	x	8.88 (±10.83)
Clinical outcomes			
Oxygen therapy, n (%)	x	x	113 (47.48)
Hospitalization >10 days, n (%)	x	x	46 (19.33)
ICU, n (%)	x	x	30 (12.61)
Death, n (%)	x	x	22 (9.24)
Hospitalization >10 days/ICU/death/oxygen therapy, n (%)	x	x	128 (53.78)

**Table 2 cells-11-01034-t002:** Algorithm performance for technical and clinical metrics and comparison of algorithm performance and inter-and intraobserver measure reproducibility’s. SD: standard deviation; Peri: pericardial volume; EAT: epicardial adipose tissue volume; MAE: mean absolute error; Corr.: correlation; O1a: initial segmentations made by Observer 1; O1b: second segmentations made by Observer 1; O2: segmentation made by Observer 2.

	O1a vs. Auto (n = 20)	O1a vs. O1b (n = 20)	O1a vs. O2 (n = 20)
**Technical Metrics**			
Peri			
Mean Dice	0.93 (±0.03)	0.92 (±0.02)	0.93 (±0.02)
Median Dice	0.93	0.92	0.93
EAT			
Mean Dice	0.85 (±0.05)	0.85 (±0.04)	0.86 (±0.03)
Median Dice	0.87	0.85	0.86
**Clinical Metric: Volume**			
Peri			
MAE (cm^3^) (mean ± SD)	35.4 (±23.4)	37.2 (±23.2)	40.3 (±22.3)
Bias (cm^3^) (mean ± SD); *p*	−6.8 (±42.7); *p* = 0.6	−22.4 (±38.3); *p* = 0.02	−21.8 (±41.3); *p* = 0.04
Corr.	0.945	0.945	0.936
EAT			
MAE (cm^3^) (mean ± SD)	11.7 (±8.1)	12.0 (±9.1)	14.9 (±11.2)
Bias (cm^3^) (mean ± SD); *p*	−4.0 (±13.9); *p* = 0.18	−6.0 (±14.0); *p* = 0.06	−12.8 (±13.6); *p* < 0.01
Corr.	0.963	0.962	0.970

**Table 3 cells-11-01034-t003:** Correlation between lung lesions extent and epicardial adipose tissue volume and epicardial adipose tissue extent (n = 238). EAT: epicardial adipose tissue volume (cm^3^); Lesion_Ext: COVID-19 lung lesion extent (%); Ext_EAT: epicardial adipose tissue extent (%).

DL_Measures	Mean (±SD)	Minimum	Median	Max	Pearson Correlation Coefficient (*p*-Value)
EAT (cm^3^)	112.8 ± 59.3	7.5	105.5	312.8	
Lesion _Ext (%)	8.9 ± 10.8	0.0	5.0	65.9	0.139 (0.037)
Ext_EAT (%)	15.6 ± 6.5	2.2	15.0	34.6	
Lesion _Ext (%)	8.9 ± 10.8	0.0	5.0	65.9	0.043 (0.522)

**Table 4 cells-11-01034-t004:** Associations with combined clinical outcome (death, ICU transfer, need for oxygen therapy, >10 days hospitalization)—Multivariable logistic regressions (n = 238). DL: Deep-learning model; Ext_EAT: epicardial adipose tissue extent (%); Lesion_Ext: COVID-19 lung lesion extent (%); ICU: intensive care unit; *: test versus “Model no DL”.

	Model No DL			Model Ext_EAT			Model Lesion_Ext			Model Ext_EAT + Lesion_Ext		
	OR	95% CI		OR	95% CI		OR	95% CI		OR	95% CI	
Gender (Male vs. Female)	2.92	1.65	5.15	2.96	1.66	5.29	2.35	1.27	4.38	2.33	1.23	4.40
Age	1.04	1.02	1.06	1.04	1.02	1.06	1.03	1.01	1.05	1.03	1.01	1.05
Number of comorbidities (1 vs. 0)	1.51	0.75	3.04	1.73	0.84	3.59	1.39	0.64	3.03	1.62	0.72	3.64
Number of comorbidities (2 vs. 0)	1.22	0.62	2.42	1.18	0.58	2.41	1.26	0.60	2.61	1.28	0.60	2.77
Ext_EAT (%)				1.04	1.00	1.09				1.05	0.99	1.10
Lesion_Ext							1.10	1.05	1.15	1.10	1.05	1.15
Area Under Curve (AUC)	0.733			0.744 (*p* * = 0.3169)			0.800 (*p* * = 0.0047)			0.805 (*p* * = 0.0029).		

## Data Availability

Not applicable.

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
