# Peer review of "Automatic Deep-Learning Segmentation of Epicardial Adipose Tissue from Low-Dose Chest CT and Prognosis Impact on COVID-19"

_cells, 2022, doi:10.3390/cells11061034_

Round 1

Reviewer 1 Report

This is a challenging study. Authors tried to demonstrate a deep learning based pipeline allowing an automated segmentation of epicardial adipose tissue (EAT) from low-dose computed tomography (LDCT) and investigate the link between EAT and COVID-19 clinical outcome. As authors pointed out, I agree that there is controversial discussion & studies regarding association between EAT and inflammatory lesions (e.g., pneumonia, myocardial damage, and adverse outcomes in COVID-19) (as shown in references: 12, 13, 14, 15, 16 in the manuscript). Also I believe it would be a great approach to develop and evaluate a deep learning pipeline that allows an automated segmentation of EAT volume in association with pre-existing COVID-19 lung lesions segmentation tool. This approach is straightforward I believe. 

However, unfortunately, authors collected not enough clinical cases for this study I think. I fully understood that the model both "Lesions_Ext" and "Ext_EAT+Lesions_Ext" reached AUC to 0.800 and they were statistically significant AUCs as compared to "No DL"; this result is good. However, AUCs were not good enough for clinically available. Authors should demonstrate results based on more cases (at least 200 training sets and 500 validation cohort I believe for this study) from various clinical institutes (at least 2 institutes).

Overall, this study is interesting and research proposal/aim is very good I think. However, authors did not investigate based on enough number of clinical data. 

Reviewer 2 Report

In the present study, the authors first develop a DL pipeline for automatic segmentation of epicardial fat in low dose chest CT (which is working in both mediastinal and lung window) by training it on 95 patients and testing it on 20 patients. Then 238 COVID-19 patients are used for the prognosis analysis, where epicardial fat segmentation and lung parenchyma lesions segmentation are used to stratify the risk of worse outcomes (death, ICU, transfer, need for oxygen therapy, >10days of hospitalization).

GENERAL COMMENTS

1/ The paper is well constructed and follows adequate rules for DL pipeline (training, testing then validation, all on different cohorts)

2/ The results are interesting and are complementary to the literature

3/ There are however too much results presented, and the readability of the paper is low. I would advise to simplify the presentation by selecting major results only and putting all other results in supplementary files.

4/ English editing is required. I have noted some typo within the PDF.

SPECIFIC COMMENTS

Title: OK

Abstract: OK

Introduction: too long. Should be narrowed and more focused on the topic of COVID.

M&M: well described but still too long. The discussion in chapter 2.3.2 does not belong to M&M. The type of computer used to run the DL model should be mentioned (to see if routinely available or limited to lab).

Results: too detailed.

Discussion: too long. Should be narrowed to more salient points.

Figures: Fig 1 gives the (false) impression that all 353 CT were included in the prognosis analysis. This should be modified.

Tables: too much tables

Ref: please consider adding this ref https://www.sciencedirect.com/science/article/abs/pii/S2211568421002254

Author Response

Response to Reviewer 2 Comments

In the present study, the authors first develop a DL pipeline for automatic segmentation of epicardial fat in low dose chest CT (which is working in both mediastinal and lung window) by training it on 95 patients and testing it on 20 patients. Then 238 COVID-19 patients are used for the prognosis analysis, where epicardial fat segmentation and lung parenchyma lesions segmentation are used to stratify the risk of worse outcomes (death, ICU, transfer, need for oxygen therapy, >10days of hospitalization).

GENERAL COMMENTS

1/ The paper is well constructed and follows adequate rules for DL pipeline (training, testing then validation, all on different cohorts)

Response: Thank you for a such positive feedback regarding our work.

2/ The results are interesting and are complementary to the literature

Response: Thank you for this valuable comment.

3/ There are however too much results presented, and the readability of the paper is low. I would advise to simplify the presentation by selecting major results only and putting all other results in supplementary files.

Response: following your advice, many results have been put in the Supplementary Files section.

4/ English editing is required. I have noted some typo within the PDF.

Response: Following your advice, the whole paper have been reviewed by a native-english editor and has been edited for proper English language, grammar, spelling, punctuation and overall style. modifications have been made in the revised version of the manuscript

SPECIFIC COMMENTS

Title: OK

Abstract: OK

Introduction: too long. Should be narrowed and more focused on the topic of COVID.

Response: Following your advice, modifications have been made in the revised version of the manuscript. Introduction  about EAT was reduced and following sentences were removed : (Page 1 – Line 37-38), (Page 1, line 44-45) (Page 2, Line 58-61) of the original mansucript and sentences have been shortened.

M&M: well described but still too long. The discussion in chapter 2.3.2 does not belong to M&M. The type of computer used to run the DL model should be mentioned (to see if routinely available or limited to lab).

Response: Following your advice, modifications have been made in the revised version of the manuscript. CT technical parameters (Page 3 – Line 112 – 115) are described in Appendix/Supplementary File section.

Results: too detailed.

Response: Following your advice, some results have been put in Appendix/Supplementary File section.

Discussion: too long. Should be narrowed to more salient points.

Response: Following your advice, modifications have been made in the revised version of the manuscript. Discussion has been divided into technical algorithm comparision (shortened – Pages 13 – 14 – Lines 291 – 213 of the original mansucript), population characteristics (shortened), algorithm performance in comparison to the litterature, discussion on EAT and COVID-19 association, limitations.

Figures: Fig 1 gives the (false) impression that all 353 CT were included in the prognosis analysis. This should be modified.

Response: Following your advice, Figure 1 was modified to better differentiate the development of the algorithm and the prognosis evaluation.

Tables: too much tables

Response: Following your advice, table 2 and 3 have been gathered and Table 5 have been put in the Appendix section 

Ref: please consider adding this ref https://www.sciencedirect.com/science/article/abs/pii/S2211568421002254

Response: Following your advice, reference has been added (Page 4; § 2.3.1 Manual Segmentation).

Reviewer 3 Report

The authors describe a machine learning method to assess the volume of epicardial adipose tissue from CT imaging; more interestingly, they suggest that this parameter could be useful in the prognosis of COVID-19 patients.

The work is scientifically robust and methodologically well conducted; it can be accepted in present form.

Author Response

Response to Reviewer 3 Comments

The authors describe a machine learning method to assess the volume of
epicardial adipose tissue from CT imaging; more interestingly, they
suggest that this parameter could be useful in the prognosis of COVID-19
patients.

The work is scientifically robust and methodologically well conducted;
it can be accepted in present form

Response: Thank you for such a positive feedback regarding our manuscript.

Round 2

Reviewer 1 Report

I understood the authors' comments. There is a limitation in this study; however, the revised manuscript is much better than initial submission, I believe.